# Using networks to analyze and visualize the distribution of overlapping genes in virus genomes

Laura Muñoz-Baena[1], Art F. Y. Poon[1,2]*

1 Department of Microbiology and Immunology, Western University, London, ON, Canada, 2 Department of Pathology and Laboratory Medicine, Western University, London, ON, Canada

* apoon42@uwo.ca

## Abstract

Gene overlap occurs when two or more genes are encoded by the same nucleotides. This phenomenon is found in all taxonomic domains, but is particularly common in viruses, where it may increase the information content of compact genomes or influence the creation of new genes. Here we report a global comparative study of overlapping open reading frames (OvRFs) of 12,609 virus reference genomes in the NCBI database. We retrieved metadata associated with all annotated open reading frames (ORFs) in each genome record to calculate the number, length, and frameshift of OvRFs. Our results show that while the number of OvRFs increases with genome length, they tend to be shorter in longer genomes. The majority of overlaps involve +2 frameshifts, predominantly found in dsDNA viruses. Antisense overlaps in which one of the ORFs was encoded in the same frame on the opposite strand (−0) tend to be longer. Next, we develop a new graph-based representation of the distribution of overlaps among the ORFs of genomes in a given virus family. In the absence of an unambiguous partition of ORFs by homology at this taxonomic level, we used an alignment-free k-mer based approach to cluster protein coding sequences by similarity. We connect these clusters with two types of directed edges to indicate (1) that constituent ORFs are adjacent in one or more genomes, and (2) that these ORFs overlap. These adjacency graphs not only provide a natural visualization scheme, but also a novel statistical framework for analyzing the effects of gene- and genome-level attributes on the frequencies of overlaps.

## Author summary

Gene overlap occurs when the same part of a genome encodes two or more genes. This phenomenon is found in all biological domains of life, but it is particularly common in viruses, where it may play a role in making viral genomes more compact. To understand the prevalence of overlapping genes in viruses, we analyzed over 12,000 genomes of every known type of virus for which this genetic information is available. Although overlaps are more abundant in viruses with larger genomes, for instance, they are also significantly shorter. Overlaps in which one of the genes is read in the opposite direction (−0 overlaps)

**Data Availability Statement:** All research data are publicly available at https://github.com/PoonLab/ovrf-viz.

**Funding:** This work was supported by a Discovery Grant from the Natural Sciences and Engineering

Research Council of Canada (05516-2018 RGPIN)
to AFYP. The funders had no role in study design,
data collection and analysis, decision to publish, or
preparation of the manuscript.

tend to be longer, which may be an emergent property of the universal genetic code. We
developed a new computational method to analyze and visualize the distribution of over-
laps among genomes belonging to a group (family) of viruses as a network. This approach
enabled us to identify distinct patterns in the organization of genomes within virus fami-
lies; for example, gene overlap in the coronavirus family tends to involve non-essential
genes outside of the "core" of the network of genes.

## Introduction

Viruses are an enormous part of the natural world, representing the majority of entities in our
planet that undergo organic evolution. For instance, a recent study estimated the existence of
over $10^{31}$ bacterial viruses, *i.e*, bacteriophage [1], which is only a fraction of viral diversity. A
particularly noteworthy feature of virus genomes is the ubiquitous presence of overlapping
reading frames (OvRFs): portions of the genome where the same nucleotide sequence encodes
more than one protein. OvRFs have been documented in all seven Baltimore classes—catego-
ries of viruses by genetic material, including double-stranded DNA (dsDNA) and positive sin-
gle-stranded RNA (ssRNA+) viruses [2]. A number of hypothetical mechanisms have been
proposed to explain this abundance of OvRFs in viruses. First, the prevalence of overlapping
genes is hypothesized to be related to genome size. Given that genomes of many viruses are
physically constrained by capsid size [3], OvRFs provide a mechanism for encoding more
information in a given genome length. Another model proposes that OvRFs could be also used
by viruses as a mechanism to accommodate high mutation rates by amplifying the effect size
of deleterious mutations (antiredundancy), such that purifying selection removes these muta-
tions more efficiently from the population [4, 5]. In addition, OvRFs have been suggested to be
a symptom of gene origination, where a new open reading frame (ORF) may arise within the
transcriptional context of an existing ORF [6]. Recent studies have produced comparative evi-
dence that these *de novo* genes will not initially have a well-established function, but will be
able to acquire it over time [7].

Previously, Schlub and Holmes [8] analyzed overlapping genes in 7,450 reference virus
genomes in the NCBI viral genomes database [9] to confirm that the number of OvRFs per
genome, as well as the number of bases within OvRFs, increases significantly with genome
length. In contrast with previous research, however, they also reported that this association
was more pronounced in DNA viruses than RNA viruses, and in double-stranded versus sin-
gle-stranded genomes. Like related work in the literature [3, 5], their comparative study
employed quantities like the number of OvRFs or total overlap length (*i.e.*, the number of
nucleotides in overlapping regions) that do not distinguish one ORF from another. In other
words, these are summary statistics where the entire genome is the unit of observation.

Our objective is to incorporate gene homology into characterizing the distribution of
OvRFs in virus genomes, with the intent of gaining a more detailed understanding of this phe-
nomenon. This comparative analysis relies on accurate annotation of ORFs in reference
genomes. Gene annotation is an increasingly challenging problem, however. For instance, the
number of reference virus genomes in the NCBI RefSeq database increased more than five-
fold between 2000 and 2015, driven in part by the increasing use of next-generation sequenc-
ing platforms [9]. Many putative ORFs in newly discovered virus genomes have no recogniz-
able homologs in protein sequence databases [10]. Furthermore, ORFs in reference genomes
are not always annotated with consistent labels, or are assigned the wrong label altogether.
Misannotations are sufficiently prevalent that there are multiple collaborations to create and

maintain databases of specific categories of genomes with manually-curated gene annotations [11, 12]. To develop a global picture of OvRF diversity across viruses at gene-level resolution, we need an automated method to efficiently label homologous ORFs for related virus genomes.

Here we report a comparative analysis of OvRFs in 12,609 virus reference genomes in the NCBI virus database. First we use conventional genome-level summary statistics to revisit fundamental questions about OvRFs in viruses, *e.g.*, do overlaps tend to occur between certain reading frames in viruses?. Next, we develop and employ an alignment-free method for clustering ORFs by sequence homology within a given virus family. This enables us to generate graphs where nodes represent clusters of homologous ORFs. These nodes are connected by two types of edges that indicate the adjacency of ORFs in genomes and the presence of overlaps, respectively. This graph-based approach not only provides an inherent visualization method for the diversity of OvRFs among different virus families, but also enables us to access the rich library of network statistics [13] to characterize the abundance and distribution of OvRFs in virus families.

## Materials and methods

### Data collection and processing

First, we downloaded the accession list of all available virus genomes from the NCBI Viral Genomes Resource [9] (https://www.ncbi.nlm.nih.gov/genome/viruses/, accessed on 2020–09-28), a community-based effort to curate references from the growing number of virus genomes in the NCBI Genbank database. This tab-separated file comprised 247,941 rows and six columns labeled as 'representative', 'neighbor', 'host', 'taxonomy' and 'segment name'. Representative genomes are used to denote significant intra-specific variation that cannot be adequately captured by a single reference genome, whereas neighbors are additional validated and complete or nearly-complete genomes for a given species [9]. We used only a single representative genome for each species as sufficient information for our purposes. We used a Python script to retrieve additional metadata (genome length, number of proteins, topology and molecule type) associated with each reference genome using the NCBI Entrez API [14, 15].

The same script was used to generate a tabular dataset recording the genome accession number, product, strand, coordinates and start codon position for every coding sequence (CDS). A second Python script was used to identify putative overlapping open reading frames (OvRFs) from the genome coordinates of all CDSs by accession number. Every OvRF was recorded by its location, length in nucleotides, and shift (if applicable) relative to the upstream reading frame. Following convention [16], overlaps between reading frames on the same strand were recorded as +0, +1 and +2 when shifted by zero, one and two nucleotides, respectively. Similarly, overlaps on opposing strands were recorded as −0, −1 and −2 (see S1 Fig). Next, we extracted Baltimore classifications for virus families from the Swiss-Prot virus annotation resource (https://viralzone.expasy.org [17]).

### Clustering protein data by family

To analyze the distribution of overlapping open reading frames (OvRFs) in different virus families, we retrieved the protein sequences for all CDSs of all reference genomes of each family from the NCBI virus database. Our objective was to identify homology among protein coding sequences that may be highly divergent and inconsistently annotated at the family level of virus diversity. We also needed to be able to accommodate gene duplication and divergence in DNA viruses, as well as unique ORFs with no homologs in other genomes (*i.e.*, accessory genes, ORFans [18]). As a result, we decided to use an alignment-free method to compute *k*-

mer-based similarity scores between every pair of ORFs within a virus family (S3 Fig). We used Python to map each protein sequence to a dictionary of $k$-mer counts for $k = \{1, 2, 3\}$ as a compact representation of the sparse feature vector. Let $W(s)$ represent the set of all $k$-mers (words) in a sequence $s$, and let $f(s, w)$ represent the frequency of $k$-mer $w$ in $s$. Using these quantities, we calculated the Bray-Curtis distance [19] between sequences $s$ and $t$:

$$k(s, t) = 1 - \frac{\sum_{W(s) \cap W(t)} 2 \min(f(s, w), f(t, w))}{\sum_{W(s)} f(s, w) + \sum_{W(t)} f(t, w)}$$

This k-mer distance performed relatively well at the task of protein classification in a recent benchmarking study of alignment-free methods [20], where it was implemented as the intersection distance in the AFKS toolkit [21]. Intuitively, this measure reflects the overlap of two frequency distributions, normalized by the total area of each distribution. The resulting distance matrix was used as input for the t-distributed stochastic neighbor embedding (t-SNE) method implemented in the R package *Rtsne* [22]. This dimensionality reduction method embeds the data points into a lower-dimensional space in such a way that the pairwise distances are preserved as much as possible. Next, we generated a new distance matrix from the coordinates of the embedded points and then used hierarchical clustering using the R function *hclust* with Ward's criterion [23] ('ward.D2'). Combining dimensionality reduction and clustering methods is frequently used in combination because distance measures have unexpected properties in high dimensional feature spaces [24].

Finally, we used the R function *cutree* to extract clusters by applying a height cutoff to the dendrogram produced by *hclust*. Increasing the number of clusters by lowering this cutoff accommodates more ORFans. Conversely, raising the cutoff reduces the number of false positive clusters (ORFs that should not be classified as ORFans). To determine an optimal cutoff for a given virus family, we selected the height that balances two quantities. Let $f(i, j)$ be the number of ORFs assigned to cluster $j \in \{1, \ldots, K\}$ in genome $i \in \{1, \ldots, N\}$. First, we calculated the mean proportion of ORFs with unique cluster assignments per genome:

$$E_1 = \frac{1}{N} \sum_{i=1}^{N} \left( \frac{\sum_{j=1}^{K} I(f(i, j) = 1)}{\sum_{j=1}^{K} I(f(i, j) > 0)} \right)$$

where $I(x)$ is an indicator function that assumes a value of 1 if $x$ is true, and 0 otherwise.

Second, we calculated the mean frequency of a cluster assignment across genomes:

$$E_2 = \frac{1}{K} \sum_{j=1}^{K} \left( \frac{1}{N} \sum_{i=1}^{N} f(i, j) \right)$$

$E_1$ increases with an increasing number of clusters, whereas $E_2$ declines because ORFs are distributed across more clusters. Thus, we passed the squared difference $(E_1 - E_2)^2$ as an objective function for R function *optimize* to locate the optimal cutoff for each virus family.

## Data visualization

Using the OvRF coordinate data from the preceding analysis, we used a Python script to generate adjacency graphs as node and edge lists for each virus family. Every cluster of homologous ORFs is represented by a node. Each node has two 'connectors' representing the 5' and 3' ends of the corresponding ORFs in the cluster. Such coordinates are drawn from the annotated locations for each CDS. For a given genome sequence, all ORFs are sorted by the nucleotide coordinates of their 5' and 3' ends in increasing order. When genes are encoded on the complementary strand their coordinates keep an ascendant order, and the ORF is tagged with a −1

to indicate the difference in orientation so they can be processed using the same procedure. Next, we evaluate every adjacent pair of ORFs in this sorted list. If the 3' end of the first ORF occupies a higher coordinate than the 5' end of another ORF, then the pair are labelled as overlapping. After screening all adjacent pairs for overlaps, the results were serialized as a weighted graph in the Graphviz DOT language [25], where each node represents a cluster of homologous ORFs. Specifically, we generated two edge lists, one weighted by the frequency that ORFs in either cluster were adjacent in genomes, and a second weighted by the frequency of overlaps. When rendering graphs, we varied edge widths in proportion to the respective weights. In addition, we used the Matplotlib [26] library in Python to visualize the gene order (synteny) of representative genomes in each virus family as concatenated ORFs coloured by cluster assignments, and to visualize the distribution of gene labels by cluster as 'word clouds'.

## Graph analysis

To analyze the distribution of OvRFs in the context of the adjacency graph of a given virus family, we encoded the numbers of overlaps between every pair of clusters (represented by nodes) as a binomial outcome, given the weight of the corresponding adjacency edge. For every node $A$ in the graph, we recorded the number of genomes; number of adjacency edges (degree size); number of triangles ($A \leftrightarrow B \leftrightarrow C \leftrightarrow A$); transitivity (frequency of $B \leftrightarrow C$ given that the graph contains $A \leftrightarrow B$ and $A \leftrightarrow C$); and the Eigenvector centrality [27], a measure of node importance similar to Google's PageRank algorithm. Next, we summed these quantities for the two nodes of each edge. We used the resulting values as predictor variables in a zero-inflated binomial regression on the probability of overlap edges, using the *zibinomial* function in the R package VGAM [28]. This mixture model extends the binomial distribution with a third parameter for the probability of zero counts in excess of the binomial. To reduce the chance of overfitting the data, we used stepwise Akaike information criterion (AIC)-based model selection (VGAM function *step4vglm*), where the model search space was limited to the intercept-only model as the lower bound, and the full model with all predictors as the upper bound. All source code used for our analyses are available under the MIT license at https://github.com/PoonLab/ovrf-viz.

## Results

To examine the distribution of overlapping open reading frames (OvRFs) across virus genomes, we retrieved 451,228 coding sequences from 12,609 representative virus genomes as identified by the NCBI virus genomes resource [9]. Based on the annotation of coding sequences (CDS) in each record, we identified 154,687 OvRFs in 6,324 viruses (50.2%). None of the circular ssRNA viral genomes ($n = 39$) contained any OvRFs based on genome annotations. Using the taxonomic annotations in these records, we were able to assign 9,982 (79.2%) of the genomes to Baltimore groups (Fig 1B). Of the remaining 2,627 genomes, 1,001 (38.1%) were comprised of DNA and 1,626 were RNA, based on the molecular type annotation of the respective records.

## Longer genomes tend to carry shorter overlaps

As expected, the number of OvRFs per genome was positively correlated with the number ORFs (Spearman's $\rho = 0.89$, $P < 10^{-12}$, Fig 1A). In addition, the relative number of OvRFs, *i.e.*, normalized by the number of ORFs per genome, varied significantly among Baltimore groups (ANOVA, $F = 835.2$, df = 6, $P < 10^{-12}$). For instance, double-stranded DNA (dsDNA) viruses —the largest group of viruses in our sample—encode on average 202.8 ORFs and 32.7 overlaps per genome. An extreme case from the Phycodnaviridae family is the Paramecium bursaria

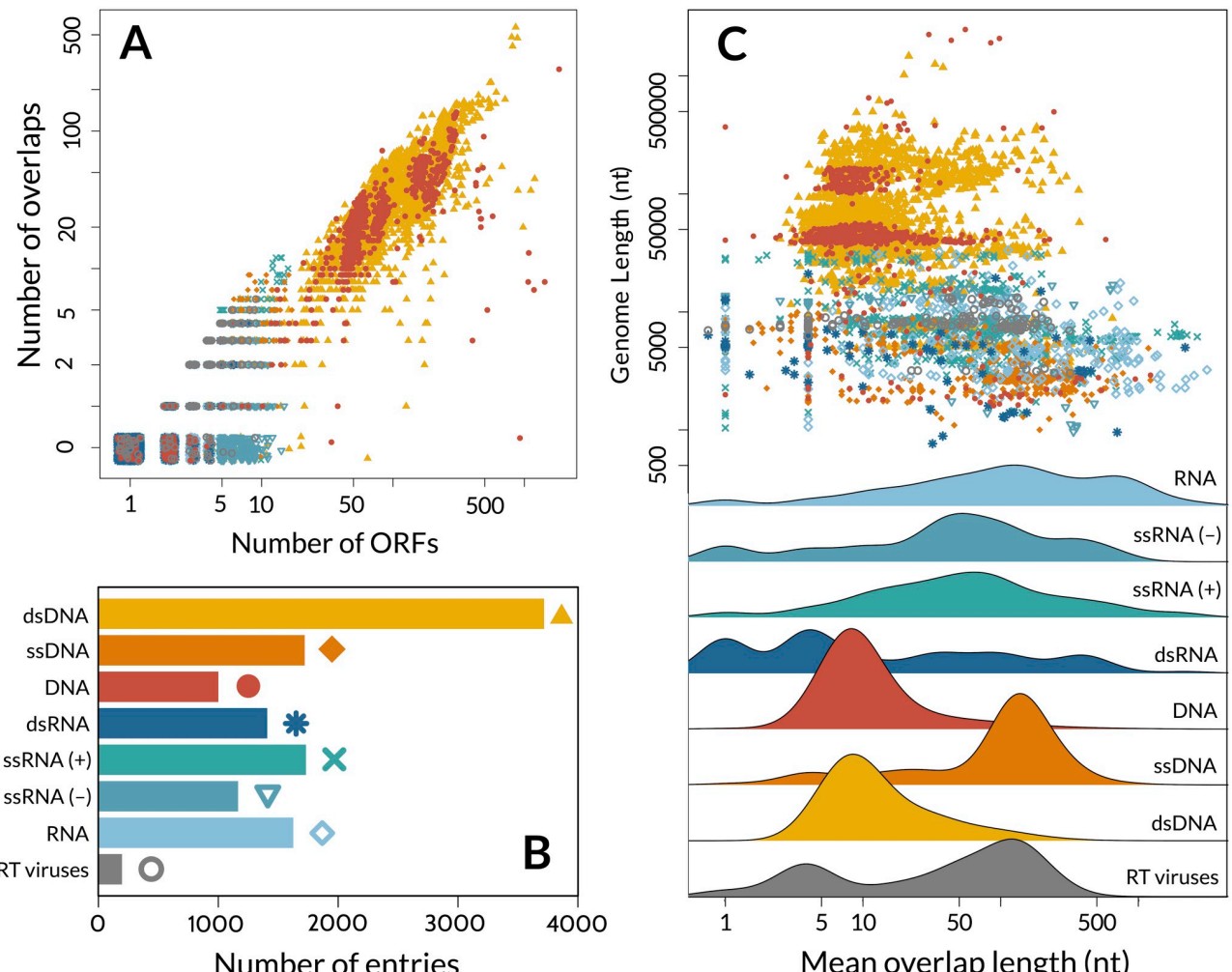

**Fig 1. Distribution of overlapping genes across virus genomes. A**. Scatterplot displaying a positive correlation between the log-transformed numbers of overlapping open reading frames (OvRFs) and ORFs per virus genome, stratified by Baltimore class. Genomes with no OvRFs were plotted at 0.5 (labeled '0') with random noise to reduce overplotting. **B**. Barplot of the number of representative virus genomes per Baltimore class, which also serves as a colour and point-type legend for the scatterplots. 'DNA' and 'RNA' correspond to the molecular type annotations of virus genomes that have not been assigned to a known virus family. **C**. A log-log scatterplot displays the distribution of genomes with respect to overall length (in nucleotides, *y*-axis) and mean length of overlapping regions (*x*-axis) by Baltimore class. Individual plots are provided in S2 Fig. Underneath, ridgeplots summarize the marginal distributions of genomes with respect to mean overlap lengths, to clarify differences between the Baltimore classes.

Chlorella virus, which infects a eukaryotic algal host and whose genome encodes 1,733 proteins with 541 (31%) overlaps. In contrast, positive sense single-stranded RNA (ssRNA+) viruses encode on average 9.2 ORFs with 2.5 overlaps per genome, and negative sense single-stranded RNA (ssRNA−) viruses encode about 7.1 ORFs and 1.6 overlaps per genome on average. Some RNA virus genomes have abundant overlapping regions, however; *e.g.*, the simian hemorrhagic fever virus genome (Genbank accession NC_003092) encodes 15 ORFs of which 10 are involved in an overlap.

In contrast, the mean number of nucleotides in overlapping regions was negatively correlated with genome length overall (Spearman's $\rho = -0.52$, $P < 10^{-12}$; Fig 1C). We note that this comparison excludes genomes without any OvRFs, which were significantly shorter (average 6038 nt versus 51424 nt; Wilcoxon rank sum test, $P < 10^{-12}$). After adjusting for multiple

comparisons ($\alpha = 6.25 \times 10^{-3}$), correlations remained significantly negative within dsDNA, dsRNA, ssRNA+ and unclassified DNA viruses only (S2 Fig). Correlations within Baltimore classes were largely driven in part by variation among virus families, and we found no consistent trend in correlations within families using a binomial test. While DNA viruses, including single-, double-stranded and unclassified species, tended to carry longer genomes (median 33489 nt, interquartile range (IQR) 2768–59073 nt), their overlapping regions tended to be relatively short (median 15.6 nt, IQR 8.5–61 nt). This trend was largely driven by the dsDNA viruses, and the distributions of overlap numbers and lengths in unclassified DNA genomes (Fig 1C) suggest that these predominantly also represent dsDNA viruses. In comparison, RNA viruses carried fewer but relatively long overlapping regions (median 169.12 nt, IQR 31.34–831.75 nt) for their shorter genome lengths (median 4046 nt, IQR 1986–8009 nt).

## Distribution of frameshifts among OvRFs

5,733 (3.7%) of the 154,687 OvRFs identified in our study involved the alternative splicing of one or both transcripts such that there is no consistent relationship between reading frames. These cases are excluded from this section because they complicate the interpretation of frame shifts. The majority ($n$ = 92,915, 62.4%) of OvRFs involved reading frames that were shifted by 2 nt on the same strand (+2; Fig 2). These mostly represented dsDNA virus genomes ($n$ = 78,191, 84.2%) and comprised almost entirely of overlaps by a single nucleotide (T[AG] ATG) or 4 nt (ATGA). (Note that the density plots in Fig 2 summarize the distribution of overlap lengths at the level of individual OvRFs, whereas Fig 1C summarizes the mean overlap lengths at the level of virus genomes. The peaks in Fig 2 do not appear in Fig 1C because a majority of 1nt and 4nt overlaps appear in a much smaller number of dsDNA genomes.) We observed +2 overlaps significantly more often among OvRFs from DNA viruses than from RNA viruses (odds ratio, OR = 3.3; Fisher's exact test, $P < 10^{-12}$). Furthermore, only four out of 29,906 (0.01%) overlaps by 1 nt involved a frame shift other than +2. These four cases involved −2 shifts where one of the ORFs was initiated by the alternate start codon TTG (*e.g.*, CATTG). Another common type of short OvRF involved −2 frame shifts with an overlap of 4 nt, *e.g.*, CTAA, where the reverse-complement of TAG is CTA. These were predominantly found in dsDNA virus genomes ($n$ = 1423, 73.3%). However, a substantial number ($n$ = 419, 21.6%) were also recorded in ssDNA viruses in which a complementary negative-sense strand is generated during virus replication, *e.g.*, Geminivirus.

Excluding OvRFs with short overlaps of 1 or 4 nt, the most common type of OvRF involved a shift of +1. These were observed in both DNA viruses ($n$ = 34,175 dsDNA, 1,448 sDNA, and 7,505 unknown) and RNA viruses ($n$ = 40 dsRNA, 62 ssRNA−, 917 ssRNA+, and 170 unknown). The median overlap length for +1 OvRFs was 14 nt (IQR 8 to 26 nt). For this type of OvRF, overlaps exceeding 2,000 nt in length were found in ssRNA+ viruses, such as Kennedya yellow mosaic virus (NC_001746) and Providence virus (NC_014126). Overlap lengths tended to be longer in association with −0 (median 114, IQR 27 − 267 nt) frameshifts (Wilcoxon rank sum test, $P < 10^{-15}$; Fig 2).

## Graph-based approach to studying OvRFs

Quantifying OvRFs by statistics like the number of overlaps per genome, or the mean overlap length, reveals substantial variation among Baltimore groups and different frameshifts. However, our objective is to characterize the distribution of OvRFs among virus genomes at a finer resolution. Specifically, these statistics, which are defined at the level of genomes, prevent us from identifying patterns in the distribution of OvRFs at the level of individual genes. For a meaningful comparison at the gene level among virus genomes, we need to be able to identify

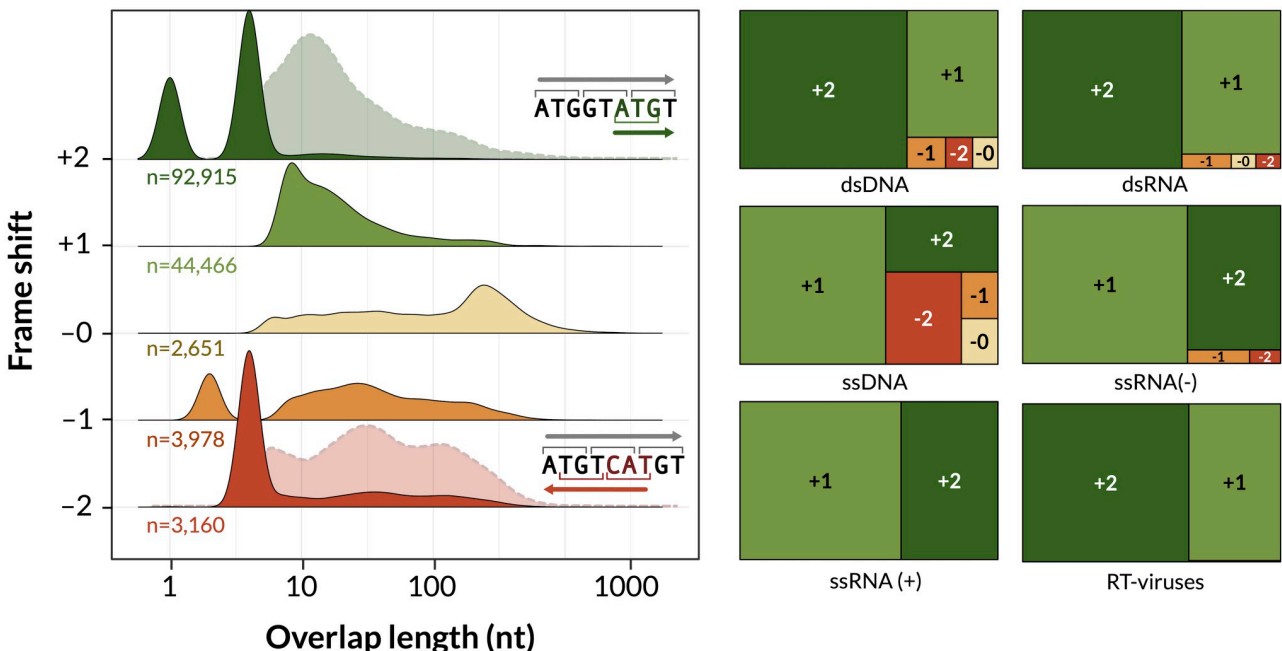

**Fig 2. Associations between overlap lengths and frame shifts.** (left) Ridgeplots summarizing the distributions of overlap lengths for different frame shifts, where +2 indicates a shift by 2 nt relative to the upstream reading frame, and −2 indicates a 2 nt shift on the opposite strand (note the reverse complement of CAT is ATG). For +2 and −2, we also display the densities after removing overlaps by 1 and 4 nt (dashed outlines), since these predominate the respective distributions. (right) Treemaps summarizing the distribution of frame shifts by Baltimore class. The area of each rectangle is scaled in approximation to the relative frequency of each frame shift.

which genes are homologs. We decided to pursue our objective at the taxonomic level of virus families, to balance diversity in OvRFs with sequence homology. Identifying homologous genes among genomes at the level of virus families is challenging, not only because of substantial evolutionary divergence, but also because genomic rearrangements that can involve the gain, loss or relocation of ORFs, *i.e.*, changes in gene order (synteny). For example, the family Rhabdoviridae is characterized for the loss and acquisition of new genes that overlap with consecutive core ORFs, driving substantial variation in genome size and the formation of new accessory genes families [29].

We used an alignment-free *k*-mer-based method [20] to partition all amino acid sequences from genomes in a given virus family into clusters of homology (S3 Fig). In brief, for each virus family we calculated a *k*-mer distance [19] between every pair of amino acid sequences. We projected the resulting distance matrix into two dimensions by t-distributed stochastic neighbor embedding (t-SNE), and then applied hierarchical clustering to the distances in the 2D plane (Fig 3A). The clustering threshold was determined by balancing the mean frequency of a cluster across genomes against the mean number of unique clusters per genome (Fig 3B).

We propose a graph-based approach to characterize the distribution of OvRFs in the context of coding sequences in the genome. This approach provides a framework for quantifying overlapping regions at a finer resolution within virus families, and is a natural method for visualizing differences between them. Each node in the graph corresponds to a cluster of homologous coding sequences. Nodes are connected by two sets of directed edges (arrows; Fig 3C). The first set represent the number of genomes in which coding sequences in the respective clusters are located next to each other (adjacency edges). A second set of edges represent the number of genomes in which the adjacent sequences are overlapping (overlap edges). Hence,

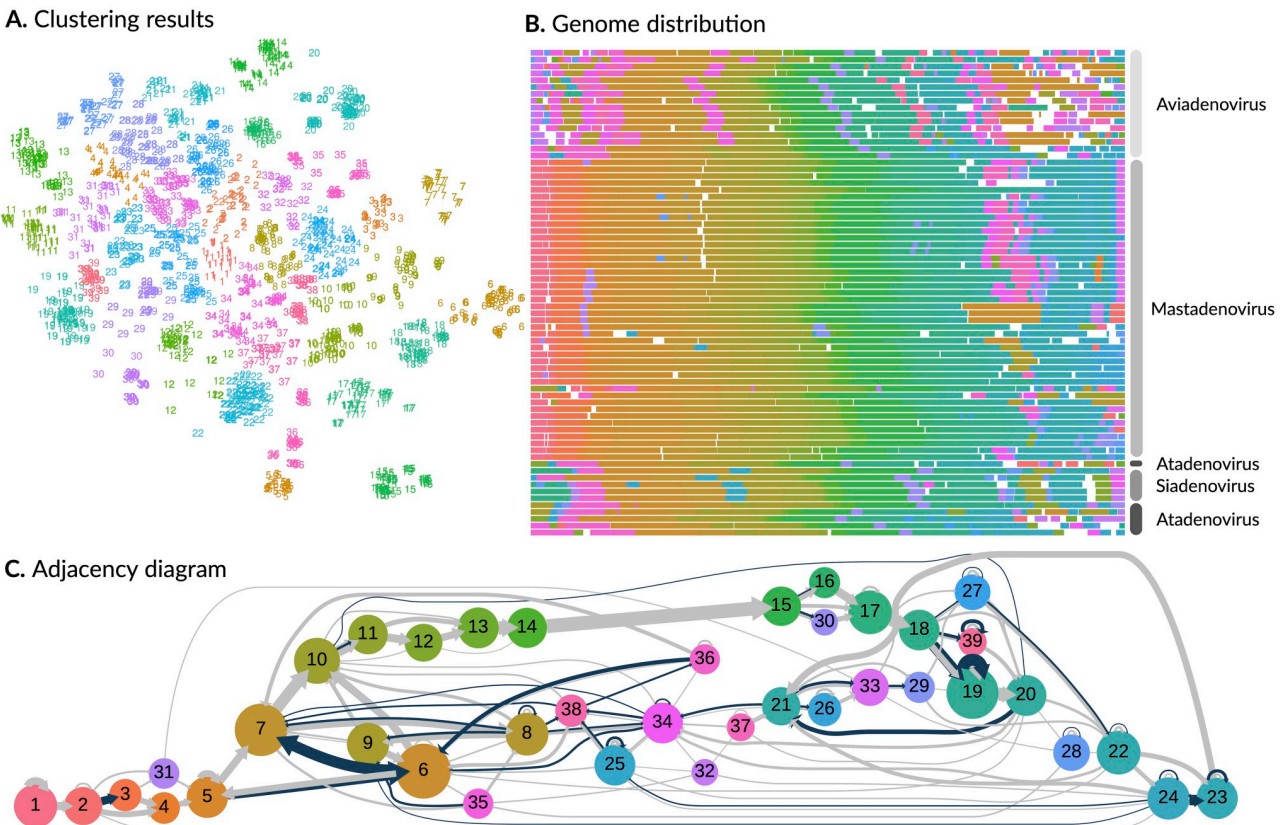

**Fig 3. Adenoviridae family analysis. A**. t-SNE projection of protein sequences from *n* = 71 genomes in the Adenoviridae. Each point represents a protein sequence, coloured and numbered by its cluster assignment. Based on our clustering criteria, we identified a total of 39 clusters for this virus family. **B**. A compact representation of reference genomes labeled by genus. Each set of line segments represent the coding sequences of a genome, coloured by cluster assignments and rescaled to a constant total length. White spaces represent non-coding regions. **C**. A hierarchical layout of the adjacency graph for Adenoviridae. Each node represents a cluster of homologous coding sequences, scaled in proportion to the number of sequences in the cluster. Node numbering and colours were determined by the order of appearance of clusters in the data. Directed edges (arrows) connect nodes representing coding sequences that are adjacent in five or more genomes. Edges are coloured blue if the genes overlap and grey otherwise; widths are scaled in proportion to the number of genomes in either case. This diagram was generated using Graphviz and arrows were manually modified in Inkscape.

an overlap edge is never present without an adjacency edge. Because edges are weighted by the number of genomes they each represent, an overlap edge can never have a weight that exceeds the matching adjacency edge.

## Example: Graph-based analysis of Adenoviridae

Adenoviridae is a family of dsDNA viruses with genomes approximately 32,000 nts in length encoding around 30 proteins. Our clustering analysis of protein sequences in the *n* = 72 reference genomes identified 37 clusters (Fig 3A). Fig 3B displays the distribution of cluster assignments across coding sequences in the genomes. We noted that one of the genomes (bovine adenovirus type 2, NC_002513) had an unusually long non-coding region. We subsequently determined that this reference genome record was not completely annotated, removed it from the dataset and repeated our analysis, resulting in 39 clusters.

The adjacency graph for Adenoviridae features a relatively conserved gene ordering that corresponds to clusters 10 to 20 (Fig 3C). In other words, this part of the graph has a mostly

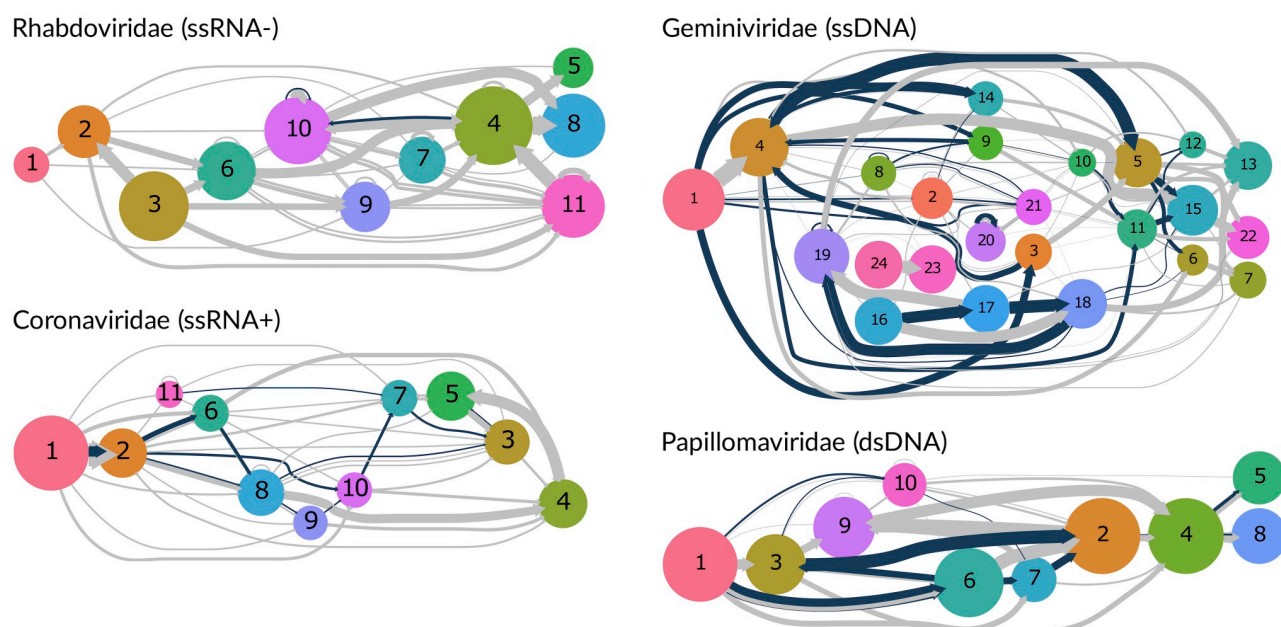

**Fig 4. Adjacency graphs for different virus families.** These graphs were generated from the clustered ORF data, using the same procedure that we employed to generate Fig 3 for Adenoviridiae. Blue edges indicate overlapping open reading frames, and grey edges represent ORFs that are adjacent but not overlapping. Edge widths were rescaled by a factor of 0.5 for Geminiviridae and Papillomaviridae to accommodate differences in sample size (numbers of genomes) among virus families. Arrowheads were manually adjusted in Inkscape as in Fig 3.

linear structure where nodes tend to have one incoming edge and one outgoing edge. Clusters 10 to 20 correspond to proteins encoded by regions L1-L5 (S4 Fig). For example, cluster 15 predominately maps to the protein names including the term 'hexon'. The graph also features several 'bubbles' in which one of the coding sequences is gained or lost in a substantial number of genomes. For example, some genomes proceed directly from cluster 11 (pVII) to 13 (pX), bypassing 12 (pV). Similarly, cluster 31 (IX or ORF0) is gained in at least 11 genomes. In addition, the graph splits between clusters 19 and 39 as it traverses from cluster 18 to 20. These clusters correspond to mixtures of the proteins 33K and 22K, which are both encoded by alternative splices of the same gene transcript that includes a long intron [30]. This split, as well as self-loop edges on both clusters, implies that our clustering method can be confounded by inconsistent annotation of such isoforms.

The graph also contains distinct groups of nodes with multiple incoming or outgoing edges, which represent homologous clusters of coding sequences with more variable orderings in Adenoviridae. For example, clusters 8, 34, 36 and 38 generally correspond to ORFs in the E1 region of aviadenovirus (genus of bird-associated adenovirus) genomes associated with gene duplication events [31]. Similarly, cluster 25 maps to the RH family of duplicated genes in aviadenovirus and atadenovirus genomes. The presence of homologous coding sequences with potentially common origins in both the E1 (5') and E4 (3') regions of the Adenoviridae genomes induces the overall cyclic structure in this adjacency graph.

In Adenoviridae, OvRFs vary from 1 to 19 nt in length, with a median length of 10 nt (IQR 8 − 12 nt). By visualizing clusters of homologous coding sequences as a graph, we can see that the conserved 'backbone' of L1-L5 genes are relatively free of overlaps. In addition, Fig 4 displays the adjacency graphs produced for four other virus families (Rhabodiviridae, Geminiviridae, Coronaviridae and Papillomaviridae, representing four different Baltimore groups. This

visual comparison of adjacency graphs not only clarifies the substantial variation in the frequency of OvRFs among families, but also reveals differences in the distribution of overlaps among ORFs. For example, OvRFs in genomes of Geminiviridae tended to be associated with common adjacency edges on the 'left' side of the graph, corresponding to homologous ORFs closer to the 5' end of genomes. OvRFs in Coronaviridae genomes tended to be associated with less common pairs of adjacent ORFs (*e.g.*, 10–7 and 7–3 versus 8–4 and 4–5; Fig 4).

## Variation among families

To analyze the distributions of OvRFs in the context of adjacency graphs, we fit a zero-inflated binomial regression model to the weights of overlap and adjacency edges for every pair of clusters. For example, out of 69 genomes with adjacent coding sequences assigned to clusters 6 and 7 in the Adenoviridae graph, 57 genomes had an overlap between the sequences and 12 did not. We calculated the number of genomes, degree size, number of triangles, transitivity and Eigenvector centrality as edge-level attributes, and used a stepwise model selection procedure to determine which combination of attributes was best supported by the data as predictor variables. The results of fitting these regression models to each graph are summarized in Fig 5.

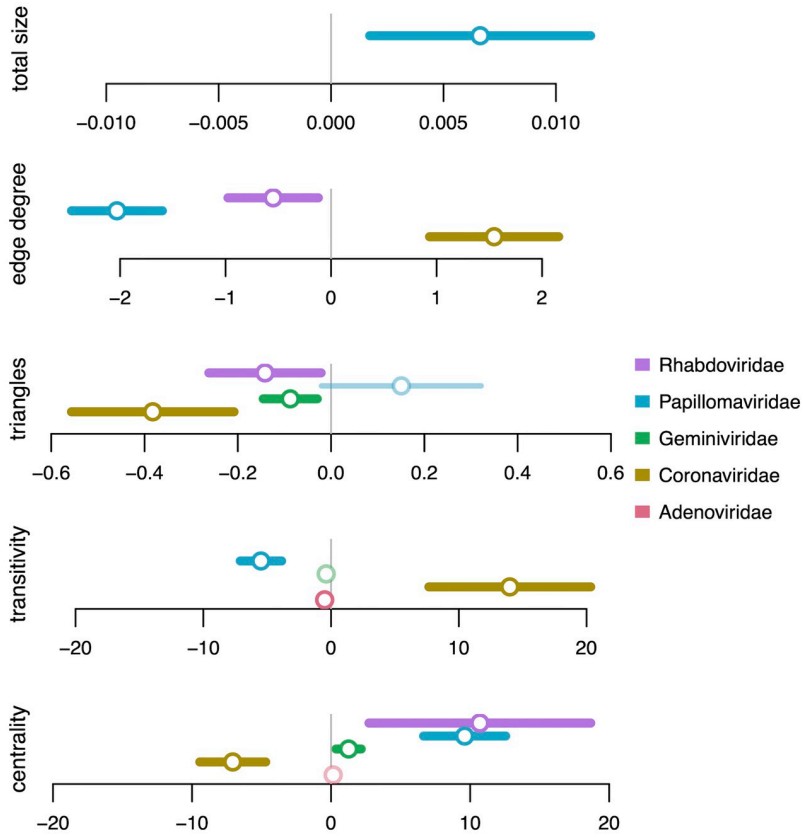

**Fig 5. Forest plot of zero-inflated binomial regressions on adjacency graphs.** Points and lines are omitted for terms that were discarded by a stepwise AIC model selection procedure. Each point corresponds to coefficient estimates from zero-inflated binomial regressions on the probability of an overlap between adjacent ORFs, given the clusters of homologous ORFs from genomes of each of five virus families (see colour legend). Line segments correspond to the 95% confidence interval of the estimate, drawn in bold when the interval does not include zero. Total size = total number of genomes with adjacent ORFs assigned to the respective clusters. Edge degree = total edge degree of the linked clusters. Triangles = total number of triangles involving either cluster. Transitivity = total transitivity of the linked clusters. Centrality = total Eigenvector centrality of the linked clusters.

Effect size estimates varied substantially among virus families. For example, centrality was significantly associated with higher probabilities of overlaps for Geminiviridae, Papillomaviridae and Rhabdoviridae, but with lower probabilities in Coronaviridae. A cluster with high centrality is connected to many other clusters that are also of high degree size. In our context, high degree sizes correspond to ORFs with variable neighbours or diverse locations in a genome, *e.g.*, due to multiple gene capture and duplication events [31]. Triangles in the adjacency graph tended to be associated with lower rates of overlap. For instance, gene loss by deletion (from $A \rightarrow B \rightarrow C$ to $A \rightarrow C$) is more likely to be tolerated if the adjacent ORFs *A* and *C* do not overlap with the targeted gene *B*. For Adenoviridae, transitivity had a relatively slight but significant negative effect on overlap probability (Fig 5). This is consistent with our previous observation that clusters comprising a core 'backbone' in the adjacency graph tended to be associated with fewer overlap edges.

## Discussion

The number of genome sequences for previously unknown viruses is rapidly accumulating in public databases, driven in part by environmental metagenomic sequencing projects [32] and education/outreach programs like SEA-PHAGES [33]. These data provide a significant opportunity to examine the composition of these genomes to identify large-scale patterns in features such as overlapping open reading frames (OvRFs). The gold standard for collecting and processing such data for comparative analyses is manual curation, which enables investigators to correct frequent misannotations in public databases, even after additional curation by collaborative efforts [9]. For example, Pavesi *et al.* [34] curated an experimentally-validated set of 80 overlapping genes from virus genomes to examine differences in nucleotide and amino acid composition from non-overlapping ORFs. Rancurel *et al.* [35] manually curated 1,098 virus genomes to identify OvRFs for a phylogenetic analysis of their role in the *de novo* emergence of novel genes. However, the scale and rate of growth of these data makes it increasingly difficult to manually curate OvRFs, and it will eventually be unfeasible to curate the full repertoire of virus genomes. While we have focused on NCBI reference genomes that have been curated through community-based efforts [9], implemented automated quality control steps (such as excluding overlaps with lengths inconsistent with other annotation) and manually inspected outliers, we recognize that the resulting database is not as reliable as those obtained through additional expert curation.

Here, our focus was on developing and applying computational methods that can scale with the rapidly growing number of genomes. We have characterized the distribution of putative OvRFs in over 12,600 annotated virus genomes. Beginning with conventional comparative methods, we first confirmed previous findings that overlapping genes are ubiquitous across all Baltimore classes, with examples identified in 50.2% of the virus genomes. We observed that the majority of non-splicing OvRFs are short (*e.g.*, less than 10 nucleotides). However, the small overlaps in our study were predominantly by 1 or 4 nucleotides, whereas previous work [8] reported peaks at slightly longer lengths (3 and 7 nucleotides, respectively). We also confirmed previous reports [2, 3, 5] that the number of OvRFs increases with genome length, whereas OvRFs tend to be shorter in longer genomes. These trends are consistent with the compression theory that proposes that overlapping genes are a significant mechanism for reducing genome lengths [3]. However, we must be cautious about interpreting these patterns because, like previous work, there is no adjustment for non-independence among observations due to evolutionary homology, *i.e.*, identity by descent. This can be mitigated in part by examining correlations within virus families. At this level, we did not find significant evidence of a consistent association between overlap and genome lengths (S2 Fig). To assess the sensitivity

of our results to misannotations that were not removed by our filtering criteria, we ran a simple simulation experiment by modifying the start and end coordinates of ORFs with random deviates from a uniform distribution $U(-10, +10)$ for 10% and 50% of all genomes, respectively, and repeating our analyses (S5 and S6 Figs). Genome-level trends in the numbers and mean lengths of overlaps were generally robust to this type and extent of misannotation. Misannotations had a more substantial impact on the distribution of frameshifts among overlaps; for instance, spurious overlaps of 1 or 2 nt became appreciably associated with −2 and +1 frameshifts with 50% misannotation; however, the overall trends remained the same.

In the absence of a standard notation for frameshifts in OvRFs, past studies have devised different labeling systems (S1 Fig). For OvRFs on the same strand, we labeled the frameshift relative to the 'upstream' reading frame. Following [16], we used a negative sign to indicate that overlaps involve ORFs on opposite strands. However, we used −1 and −2 to indicate that the codons on the opposite strand are shifted by one and two nucleotides, respectively, relative to the −0 frame, which we consider to be a more intuitive notation. In an analysis of 701 RNA virus genomes, Belshaw and collaborators [3] previously reported that most overlapping genes consist of +1 and +2 frameshifts (+1 and −1 in their notation). We observed similar results in our analysis of 5,972 RNA virus genomes, which we further stratified by Baltimore class (Fig 2). We also encountered apparent overlaps between genes in a common reading frame, which we denote as +0. Even though these OvRFs share codons, they yield different gene products where one is truncated relative to another, which may influence the folding and maturation of the respective proteins. These cases may thereby provide a means of differentiating between the compression [3] and antiredundancy [4] hypotheses of OvRFs in viruses, since +0 overlaps increase the selective burden of the same nucleotides, whereas other frameshifts increase the number of nucleotide sites under purifying selection. On the other hand, +0 overlaps have a much narrower repertoire of protein sequences and structures. Since these cases do not represent true OvRFs, we excluded them from our analysis.

In our data, antisense frameshifts (*i.e.*, −0, −1 and −2) account for only the 6.6% of all overlaps and are primarily found in DNA virus genomes. For example, we detected a total of 14 cases of antisense frameshifts in RNA virus genomes (two instances in −2, four in −1, and one in −0). For example, two −1 overlaps 434 and 44 nt in length are annotated in segment S of the dsRNA virus Pseudomonas phage phiYY (NC_042073). Since these involve hypothetical proteins in a recently discovered dsRNA phage, however, we must be cautious about interpreting these results. In DNA viruses, −1 are slightly more common than −0 and −2 OvRFs (Fig 2), especially if we exclude the most common −2 overlap by four nucleotides (*i.e.*, CTA). However, Lébre and Gascuel [16] recently determined that the −1 frameshift (−2 in their notation) was the most constrained, in that the codons used in one ORF limit the amino acids that can be encoded in the other. It also minimizes the expected frequency of stop codons in the opposing strand, but −1 overlaps were not significantly longer than other types in our data. Moreover, we observed that overlaps of frameshift −0 tended to be longer than the other antisense overlaps (Fig 2). A unique property of the −0 shift is that there any combination of amino acids can be encoded without inducing a stop codon in the reading frame opposite [16], due to redundancy in the universal genetic code. Carrying over an example from Lèbre and Gascuel, there is no way to encode two tyrosines (YY) without introducing a stop codon in the −2 reading frame. This property of −0 overlaps may play a significant role in permitting greater lengths.

One of the key challenges for extending our comparative analysis to the level of individual ORFs was the assignment of ORFs into clusters of homology. This is complicated not only by extensive sequence divergence at the level of virus families, but also the gain or loss of ORFs in different lineages through processes that include gene duplication. Furthermore, the annotation of ORFs in a general purpose public database like Genbank is not sufficiently consistent to

rely on these labels. For example, cluster 15 in our analysis of Adenoviridae genomes comprised coding sequences with diverse labels, including 'hexon', 'hexon protein', 'hexon capsid protein', 'L3 hexon', 'II', 'capsid protein II', 'protein II', and the ubiquitous 'hypothetical protein' label (S4 Fig). We also found several examples of genomes in the NCBI Viral Genome Resource in which ORFs were incompletely annotated. The bovine adenovirus type 2 reference genome (NC_002513), for example, has only 11 annotated coding sequences. Adenovirus genomes typically contain about 30 to 40 genes. Since this reference genome lacks coding sequence annotations over a 15 kbp interval, it was apparent that many genes were simply not annotated. We subsequently confirmed this using a gene prediction and homology search analysis and discarded this reference genome from our analysis.

Given the abundance of genomic diversity at the level of virus families, the assignment of ORFs into homologous groups is not unambiguous. Thus, we utilized an alignment-free approach to cluster the coding sequences in the reference genomes for each virus family. There are a large number of alignment-free methods that extract k-mers from two input sequences (see [20] for a recent review). We chose the Bray-Curtis distance (also known as the intersection distance) because it performed comparatively well at the task of protein classification in a recent benchmarking study [20]. However, the classification analysis in that study was performed on protein databases curated to span a broad range of relationships, including both cases of evolutionary and structural homology. While alignment-free methods are generally regarded as more suitable for comparing more divergent viral genomes [36], we note that it is feasible to use a conventional sequence alignment program to generate a pairwise distance matrix for clustering analysis. To illustrate, we generated a distance matrix for the Papillomaviridae genome set by pairwise alignment with MAFFT, and then applied the rest of our analysis to the result. Originally, we obtained 10 clusters using our alignment-free, k-mer-based approach to generate a distance matrix (Fig 4). Using a word cloud to visualize the distribution of labels among clusters, we observed a clear separation of E1, E2, E4, E7 and L2 labels among clusters, suggesting that the k-mer method does a reasonable job clustering homologous proteins for Papillomaviridae (S7 Fig). For example, cluster 2 is predominantly associated with 'L2' and 'minor capsid protein' labels. Applying the same threshold criterion to the p-distance matrix from pairwise alignment resulted in a similar number of clusters (9). Clusters in this graph each presented an assortment of different labels for proteins that was not consistent with homology (S8 Fig). For example, cluster 3 carried labels for L1, L2 and E1 at similar frequencies. The adjacency graph from the pairwise alignment method (S9 Fig) was topologically similar to the graph in Fig 4. For instance, overlap edges were predominantly associated with a subset of four to five nodes in both cases. However, the lack of a consistent association of labels with nodes in the alignment-based graph precluded a more direct comparison between topologies.

To extract putative clusters of homologous ORFs, we were required to define some threshold to apply to the hierarchical clustering results. For any given threshold, some number of ORFs will be misclassified into separate clusters (false negatives) or into the same cluster (false positives)—this issue is common to all unsupervised clustering methods. For each virus family, we selected thresholds that minimized the number of duplicate cluster assignments per genome, while maximizing the overall frequencies of clusters across genomes. This criterion assumes that homologous ORFs are represented by a single member in every genome. For instance, viruses in the family Coronaviridae are characterized by five conserved genes—replicase polyprotein (1ab), spike (S), envelope (E), membrane (M) and nucleocapsid (N)—and a varying number of accessory proteins of low homology [37]. The conserved gene order is reproduced in the corresponding graph (Fig 4), which comprises a distinct chain of five nodes mapping to the respective genes (S10 Fig). On the other hand, the remaining six nodes do not

readily map to consistent subsets of accessory gene labels, *e.g.*, NS3C, NS7 protein. Although it is intuitive, our criterion is as *ad hoc* approach that is not supported by an underlying model of genome evolution. Hence, developing improved and efficient clustering methods in this context will be an important area for further work.

To evaluate the role of OvRFs in the emergence of genes *de novo* by overprinting [5, 6], we need to characterize the distribution of overlaps at the level of individual ORFs. We employed unsupervised clustering of homologous ORFs to identify gene order polymorphisms at the level of virus families. Visualizing the syntenic relationships among clusters at this taxonomic level provided some interesting patterns. For example, members of Adenoviridae have about 16 conserved 'core' genes in the middle of the genome that are responsible for DNA replication and encapsidation, and the formation and structure of the virion [31]. These core genes formed a distinct backbone in our adjacency graph of this family that was relatively free of overlaps (Fig 3). However, clusters 5, 6 and 7 were connected by wide overlap edges indicating an abundance of overlaps between 5–6 and 6–7. These clusters correspond to highly conserved ORFs that were predominantly annotated as encoding the conserved maturation protein IVa2, DNA polymerase and pre-terminal protein (pTP), respectively. Thus, in some cases conservation of gene order is accompanied by conserved overlaps. On the other hand, cluster assignments tended to become more variable towards the 5' and 3' ends of the genome in association with an increasing frequency of OvRFs. Similarly, OvRFs in Coronaviridae tended to be associated with pairs of clusters representing accessory genes that were less frequently adjacent in these genomes (Fig 4). As a member of the virus order nidovirales, coronaviruses have undergone extensive selection to expand the repertoire of genes encoded by their relatively long RNA genomes [38]. Accessory genes assigned to clusters with overlaps tended to be found in varying locations in genomes of Coronaviridae, which is more consistent with horizontal gene transfer or duplication than overprinting.

Like any visualization method, there is a practical upper limit to the amount of information that can be represented by an adjacency graph. For instance, we restricted our graphs to employing node colour and size, and edge colour and width, to represent cluster identities, numbers of ORFs, and the type (adjacency, overlap) and frequency of relationships, respectively. While there is a larger repertoire of visual channels, *e.g.*, node shape, adding information would make the graph increasingly difficult to interpret. Nevertheless, there are several other attributes of overlaps that are potentially of interest, such as the average length of overlaps or the predominant frameshift. To illustrate, we provide an alternative rendition of the adjacency graph for the Adenoviridae family in which edge widths represent the mean lengths of overlaps (S11 Fig).

In summation, we have described and demonstrated a new approach to characterize the distribution of OvRF in diverse virus genomes. Adjacency graphs provide a framework for both visualizing these distributions and for hypothesis testing, *i.e.*, effects of gene- or genome-level attributes on the frequencies of overlaps between specific clusters of homologous ORFs. In future work, we will develop comparative methods on the topologies and features of adjacency graphs to identify shared characteristics between virus families at this level. We further postulate that adjacency graphs may provide useful material for extending methods for ancestral gene order reconstruction [39], where the graphs can address the problem of uncertain labelling of genes. Ideally, one would simultaneously reconstruct the phylogeny relating observed genomes. Reconstructing ancestral gene order is already an NP-hard problem [40]. Given the diverse and evolutionarily fluid composition of many virus genomes, however, it is remarkable that the gain and loss of ORFs has not been explored as much as larger organismal genomes.

## Supporting information

**S1 Fig. Notation of the 6 possible frameshifts used in this study and two other studies [3, 16].**
(TIFF)

**S2 Fig. Scatterplots of genome length on mean overlap length by Baltimore class.** Each point represents a virus genome, coloured by virus family. Families represented by only one genome were coloured in grey (smaller point size). Results from Spearman rank correlations are summarized in the lower left corner of each plot. The lower-right panel displays a box-and-whisker plot summarizing the distributions of Spearman's rank correlation coefficients ($\rho$) for genomes within families, grouped by Baltimore class.
(TIFF)

**S3 Fig. Creation of adjacency plots. A**. Steps used to generate the input file for the adjacency plot. First, we downloaded a multifasta file containing the protein sequences of the reference genomes for each species in the virus family. Then, we used a Python script to calculate the *k-mer* distance between proteins followed by an R script to designate each protein to a cluster according to homology. Finally, we used a Python script to generate dot files using Graphviz. **B**. Adjacency plot interpretation. Each one of the proteins that constitute a genome is assigned to a different cluster with homologous proteins from other species. From each cluster, we draw arrows in gray that represent adjacent proteins and arrows in blue that represent overlapping proteins. The width of the arrow is proportional to the number of proteins related between the two clusters. One cluster can have entries adjacent to proteins in different clusters. In this example, cluster 3 has proteins adjacent to proteins in cluster 2 and cluster 5.
(TIFF)

**S4 Fig. Word clouds of protein names mapped to clusters for Adenoviridae.** The size of each word (gene annotation) is scaled in proportion to its relative frequency in association with ORFs in the respective clusters.
(TIFF)

**S5 Fig. Figures regenerated after random introduction of 10% missanotation on the ORFs database.**
(TIFF)

**S6 Fig. Figures regenerated after random introduction of 50% missanotation on the ORFs database.**
(TIFF)

**S7 Fig. Distribution of protein names across clusters formed based on the analysis of distance matrix for Papillomaviridae proteins.**
(TIFF)

**S8 Fig. Distribution of protein names across clusters formed based on the analysis of a pairwise alignment of Papillomaviridae proteins.**
(TIFF)

**S9 Fig. Adjacency graph based on proteins clustered according to a pairwise alignment for Papillomaviridae.**
(TIFF)

**S10 Fig. Distribution of protein names across clusters formed based on the analysis of distance matrix for Coronaviridae proteins.**
(TIFF)

**S11 Fig. Adjacency graph of Adenoviridae family with overlapping edges proportional to overlap length.**
(TIFF)

## Author Contributions

**Conceptualization:** Art F. Y. Poon.

**Data curation:** Laura Muñoz-Baena.

**Formal analysis:** Laura Muñoz-Baena.

**Funding acquisition:** Art F. Y. Poon.

**Investigation:** Laura Muñoz-Baena.

**Methodology:** Laura Muñoz-Baena, Art F. Y. Poon.

**Project administration:** Art F. Y. Poon.

**Software:** Laura Muñoz-Baena.

**Supervision:** Art F. Y. Poon.

**Validation:** Laura Muñoz-Baena.

**Visualization:** Laura Muñoz-Baena, Art F. Y. Poon.

**Writing – original draft:** Laura Muñoz-Baena, Art F. Y. Poon.

**Writing – review & editing:** Laura Muñoz-Baena, Art F. Y. Poon.

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
