## [Decision Letter · Decision Letter 0]

15 Sep 2021

Dear Dr. Poon,

Thank you very much for submitting your manuscript "Using networks to analyze and visualize the distribution of overlapping reading frames in virus genomes" for consideration at PLOS Pathogens. As with all papers reviewed by the journal, your manuscript was reviewed by members of the editorial board and by several independent reviewers. In light of the reviews (below this email), we would like to invite the resubmission of a significantly-revised version that takes into account the reviewers' comments.

Dear Dr Poon,

you manuscript on "Using networks to analyze and visualize the distribution of overlapping reading frames in virus genomes" has now been seen by two expert reviewers. As you will see from the reviews, both reviewers recognized the importance and interest of the topic, but both had multiple, substantial critical comments, reviewer 1 being highly skeptical of the validity and impact of the work in its present form. I share the concerns of the reviewer, most especially, with respect to comments #1 and #2 of reviewer 1 that put into question the robustness of the entire approach. I will consider a thoroughly revised manuscript, but please, note that the comments of the reviewers must be addressed in full, which is expected to require extensive reanalysis of the primary data.

Please, accept my apologies for the substantial delay with the review: it is difficult to recruit reviewers, especially during the summer.

Sincerely,

Eugene Koonin

We cannot make any decision about publication until we have seen the revised manuscript and your response to the reviewers' comments. Your revised manuscript is also likely to be sent to reviewers for further evaluation.

Sincerely,

Eugene V Koonin

Guest Editor

PLOS Pathogens

David Wang

Section Editor

PLOS Pathogens

Kasturi Haldar

Editor-in-Chief

PLOS Pathogens

orcid.org/0000-0001-5065-158X

Michael Malim

Editor-in-Chief

PLOS Pathogens

orcid.org/0000-0002-7699-2064

Dear Dr Poon,

you manuscript on "Using networks to analyze and visualize the distribution of overlapping reading frames in virus genomes" has now been seen by two expert reviewers. As you will see from the reviews, both reviewers recognized the importance and interest of the topic, but both had multiple, substantial critical comments, reviewer 1 being highly skeptical of the validity and impact of the work in its present form. I share the concerns of the reviewer, most especially, with respect to comments #1 and #2 of reviewer 1 that put into question the robustness of the entire approach. I will consider a thoroughly revised manuscript, but please, note that the comments of the reviewers must be addressed in full, which is expected to require extensive reanalysis of the primary data.

Please, accept my apologies for the substantial delay with the review: it is difficult to recruit reviewers, especially during the summer.

Sincerely,

Eugene Koonin

Reviewer's Responses to Questions

**Part I - Summary**

Reviewer #1: This manuscript looks at overlapping coding sequences in all Baltimore classes of viruses. The manuscript introduces some novel approaches to the field with innovative and thorough statistical analyses. It is clearly written and the rationale for using different statistical methods is concisely explained which makes it an interesting read despite some difficult concepts. A new aspect is that, instead of just looking at global statistics (e.g. % dual coding nucleotides versus genome length and Baltimore class), it also looks at patterns of overlap on a gene-by-gene basis in different virus families. However it is unclear that this latter analysis has led to any substantial new insights - e.g. no actual outcome of this analysis is given in the abstract (instead the results quoted in the abstract focus on results that have been found in previous studies of overlapping genes - e.g. PMIDs 32071766 27209091 20610432 19640978 30339683 etc). As such, it is perhaps more of a Methodology paper.

Reviewer #2: This study summarizes gene overlaps in viruses in two major ways:

1. General statistics in respect to the Baltimore classification of viruses, and in respect to the frame of overlap (+0, +1, +2, -0, -1, -2).

2. Graph analyses by clustering ORFs from different genomes into homology clusters and connecting by two types of edges - adjacency and overlap.

The first approach mainly confirms previous findings, while the second approach is novel and presents a new way to describe, visualize and quantify gene overlaps in finer resolution separately for different virus families.

While the methods used for homology clustering are arbitrary, as stated by the authors as well, they do present an interesting new approach to analyze overlapping genes, in an evolutionary meaningful framework.

The approach and the results are of interest and impact to the study of virus genome evolution.

**Part II – Major Issues: Key Experiments Required for Acceptance**

Reviewer #1: 1) The authors rely on NCBI annotations of CDSs. However, these tend to be quite inaccurate (as the authors discovered for bovine adenovirus NC 002513, p20 lines 5-8). Overlapping and other non-standard genes (ribosomal frameshifting, non-AUG initiation, transcriptional slippage, alternative splicing, misannotation of an upstream in-frame AUG when in fact it is not actually used due to being upstream of a transcript start site, etc) are common in viruses but often dealt with poorly by automatic gene prediction software or indeed by manual annotators. Thus, many virus sequences have missing ORFs, while some sequences may have spurious ORFs annotated as if they were real CDSs.

A few misannotations could seriously bias results - particularly for reverse-frame overlapping genes - for example, the authors 4 reverse-frame overlaps in dsRNA viruses arise from "NC_003729" Reoviridae, "NC_042073" Cystoviridae, "NC_043677" Chrysoviridae, "NC_043678" Chrysoviridae and are likely all misannotations.

The authors should carefully curate their input dataset and/or show that their methods and conclusions are robust with respect to misannotations.

2) The authors include +0 frame overlaps. By this, they mean where the same coding sequence forms part of two different CDS annotations, e.g. in Venezuelan equine encephalitis alphavirus NC_001449, polyprotein P123 is encoded in the CDS 44-5684 and polyprotein P1234 is encoded in the CDS 44-7526 with readthrough of the stop codon at position 5682-5684. The authors count the region 44-5684 as a +0 frame overlap with the region 44-7526. The authors justification for counting this as an overlap is "Even though the reading frames share codons, they yield different gene products such that the codons are exposed to different selective environments." and "+0 overlaps increase the selective burden of the same nucleotides" (p19 lines 6-10). I find this difficult to swallow. First, the shared domains often have largely similar functions in the two products (e.g. alphavirus P123 versus the P123 domains of P1234) - cf. different-frame overlaps such as orthoreovirus sigma-1 and sigma-1s - where the two proteins have no sequence in common and hence no shared function at all. Second, all codons are subject to multiple selective pressures anyway (translation, mutational, protein function - many proteins are multifunctional); it doesn't make sense to me to single out +0 overlaps as being subject to additional selection pressures.

3) A second problem with including +0 frame overlaps is that their presense in a genome is very much subject to the whim of the annotator. As an example, the alphavirus NC_001449 has both readthrough and frameshift ORFs annotated:

45-5684 and 45-7526 (stop codon readthrough at 5682-5684)

7562-11329 and join(7562-9970,9970-10047) (ribosomal frameshift at 9970)

leading to +0 frame overlaps 45-5684 and 7562-9970. On the other hand, the alphavirus NC_023812 only has two CDSs annotated 43-7467 and 7541..11269. The stop codon read through and frameshift are there, but extra CDSs are not annotated so the authors find zero +0 frame overlaps.

Such annotation issues will bias the results as these mainly affect RNA viruses.

4) There seem to be issues with the annotation scripts. For example in the authors' supplementary overlaps.csv file, we have the following entries for NC_001449, listing the +0 frame P123/P1234 overlap in the first line, but then the frameshift ORF join(7562-9970,9970-10047), comprising a +0 frame overlap 7562-9970 and a +2 frame overlap 9970-10047, is listed as two +0 frame overlaps:

"NC_001449","non-structural polyprotein precursor P123",44,5684,1,"non-structural polyprotein precursor P1234",44,7526,1,5640,7482,5640,+0

"NC_001449","truncated polyprotein",7561,10047,1,"structural polyprotein precursor",7561,11329,1,2487,3768,2409,+0

"NC_001449","truncated polyprotein",7561,10047,1,"structural polyprotein precursor",7561,11329,1,2487,3768,78,+0

Another example is in simian hemorrhagic fever virus NC_003092, where the 678-nt +1 frame fragment (i.e. 2875-3552) of the join(210-2876,2875-3552) frameshift CDS is labelled as a +0 frame overlap.

"NC_003092","ORF1aTF polyprotein",209,3552,1,"ORF1a polyprotein",209,6527,1,3345,6318,2667,+0

"NC_003092","ORF1aTF polyprotein",209,3552,1,"ORF1a polyprotein",209,6527,1,3345,6318,678,+0

Reviewer #2: 1. In the graph analysis, homology is based on an alignment free method. how robust are the results of the graph analyses? Will similar conclusions be made with an alignment based homology method? How sensitive are the results and conclusions to the chosen cutoffs? To answer these questions it would be of value to repeat the graph analysis (at least on a small subset) with an alignment based method for homology clustering and possibly also modifying clustering cutoffs on the existing clustering and briefly discuss the impact of these changes on the results and conclusions of the analyses.

2. The graph analysis is lacking information of the overlap frame and overlap length, which are of great interest. Adding edges separate for frame and for length in the supplementary information would be of value (or at least adjusting edge width by a minimum of overlap length and discussing the impact on the resulting graph).

**Part III – Minor Issues: Editorial and Data Presentation Modifications**

Reviewer #1: While the authors have referenced previous studies such as Brandes & Linial, Chirico et al, and Schlub & Holmes, I feel that they should also reference Rancurel et al PMID 19640978 and Pavesi et al PMID 30339683, who used carefully curated datasets of overlaps rather than relying on NCBI annotation.

The authors frequently (e.g. p5 line 24, p5 line 25, etc) use the term "reading frame" when the term "open reading frame" (i.e. "ORF") would, I feel, be clearer. A genome has 6 (potential) reading frames (viz. +0,+1,+2,-0,-1,-2) but (for large genomes) can have 100s of ORFs.

On p6 lines 10-11, I feel it would be useful to state explicitly how (-)strand ORFs (where the biological 5' and 3' ends are in the opposite orientation with respect to the genome coordinates) are dealt with.

In Fig 4, why are there only 11 clusters in the coronaviridae graph, given that SARS alone has 13 ORFs and there are additional coronavirus accessory genes present in other coronaviruses?

The authors say that simian hemorrhagic fever virus genome (NC 003092) "encodes 33 reading frames" (p9 line 12). I don't know where this number came from because NC_003092 only has ~15 CDSs.

Reviewer #2: 1. There seems to be a discrepancy between figure 1c and figure 2a - in overlap of frame +2 the majority of overlap lengths is 1 and 4, and overlap of +2 is the dominant in both dsDNA and dsRNA, however in Figure 1c there are no peaks for dsDNA with length 1 and 4, this does not make sense because there are more dsDNA than dsRNA in the analysis. This apparent discrepancy needs to be fixed or well explained in the text.

2. Long overlaps with -0 frame are not explained - is this unique to a single edge between homology clusters or does this occur in several edges, if the latter is true this might have implications for the evolution of the genetic code.

3. Although most adjacent nodes don't show overlap, some large nodes have thick edges (e.g., nodes 6 and 7 in the Adenoviridae family) suggesting that in some cases adjacency of specific genes is accompanied by conserved overlap between the two genes - this should be discussed.

PLOS authors have the option to publish the peer review history of their article (what does this mean?). If published, this will include your full peer review and any attached files.

Reviewer #1: No

Reviewer #2: No
---

## [Decision Letter · Decision Letter 1]

2 Feb 2022

Dear Dr. Poon,

We are pleased to inform you that your manuscript 'Using networks to analyze and visualize the distribution of overlapping genes in virus genomes' has been provisionally accepted for publication in PLOS Pathogens.

Best regards,

Eugene V Koonin

Guest Editor

PLOS Pathogens

David Wang

Section Editor

PLOS Pathogens

Kasturi Haldar

Editor-in-Chief

PLOS Pathogens

orcid.org/0000-0001-5065-158X

Michael Malim

Editor-in-Chief

PLOS Pathogens

orcid.org/0000-0002-7699-2064

Dear Dr Poon,

Your revised manuscript on "Using networks to analyze and visualize the distribution of overlapping genes in virus genomes" has now been reviewed by the two original reviewers who both find the revision to be fully satisfactory and have no further comments.

Therefore, I am pleased to recommend acceptance of your manuscript for publication in its present form.

Sincerely,

Eugene Koonin

Reviewer Comments (if any, and for reference):

Reviewer's Responses to Questions

**Part I - Summary**

Reviewer #1: All my previously raised points have been carefully addressed by the authors.

Reviewer #2: The authors addressed all issues raised in the first revision.

**Part II – Major Issues: Key Experiments Required for Acceptance**

Reviewer #1: (No Response)

Reviewer #2: (No Response)

**Part III – Minor Issues: Editorial and Data Presentation Modifications**

Reviewer #1: (No Response)

Reviewer #2: (No Response)

PLOS authors have the option to publish the peer review history of their article (what does this mean?). If published, this will include your full peer review and any attached files.

Reviewer #1: No

Reviewer #2: No

---

## [Editor Report · Acceptance letter]

21 Feb 2022

Dear Dr. Poon,

We are delighted to inform you that your manuscript, "Using networks to analyze and visualize the distribution of overlapping genes in virus genomes," has been formally accepted for publication in PLOS Pathogens.

Best regards,

Kasturi Haldar

Editor-in-Chief

PLOS Pathogens

orcid.org/0000-0001-5065-158X

Michael Malim

Editor-in-Chief

PLOS Pathogens

orcid.org/0000-0002-7699-2064